


# Diverse Chemical Mixing States of Aerosol Particles in the Southeastern United States

Amy L. Bondy[1], Daniel Bonanno[2], Ryan C. Moffet[2], Bingbing Wang[3,a], Alexander Laskin[3,b], Andrew P. Ault[1,4]

[1]Department of Chemistry, University of Michigan, Ann Arbor, MI, 48109, USA
[2]Department of Chemistry, University of the Pacific, Stockton, CA, 95211, USA
[3]Environmental Molecular Sciences Laboratory, Pacific Northwest National Laboratory, Richland, WA, 99354, USA
[4]Department of Environmental Health Sciences, University of Michigan, Ann Arbor, MI, 48109, USA
[a]present address: State Key Laboratory of Marine Environmental Science, College of Ocean and Earth Sciences, Xiamen University, Xiamen, China
[b]present address: Department of Chemistry, Purdue University, West Lafayette, IN, 47907, USA

*Correspondence to*: Andrew P. Ault (aulta@umich.edu)

**Abstract.** Aerosols in the atmosphere are chemically complex with thousands or more chemical species distributed in different proportions across individual particles in an aerosol population. An internal mixing assumption, with species present in the same proportions across all aerosols, is used in many models and calculations of secondary organic aerosol (SOA) formation, cloud activation, and aerosol optical properties. However, many of these effects depend on the distribution of species within individual particles, and important information can be lost when internal mixtures are assumed. Herein, we show that during the Southern Oxidant and Aerosol Study (SOAS) in Centreville, Alabama, at a rural, forested location, that aerosols frequently are not purely internally mixed, even in the accumulation mode (0.2-1.0 µm). A range of aerosol sources and mixing states were obtained using computer controlled scanning electron microscopy with energy dispersive X-ray spectroscopy (CCSEM-EDX) and scanning transmission X-ray microscopy-near-edge X-ray absorption fine structure spectroscopy (STXM-NEXAFS). Particles that were dominated by SOA and inorganic salts were the majority of particles by number fraction from 0.2-5 microns with an average of 78% SOA in the accumulation mode. However, during certain periods contributions by sea spray aerosol (SSA) and mineral dust were significant to accumulation (22 % SSA and 26 % dust) and coarse mode number concentrations (38 % SSA and 63% dust). The fraction of particles containing key elements (Na, Mg, K, Ca, and Fe) were determined as a



function of size for specific classes of particles. Within internally mixed SOA/sulfate particles < 5 % contained Na, Mg, K, Ca, or Fe, though these non-volatile cations were present in particles from the other sources (e.g. SSA and dust). Mass estimates of the aerosol elemental components were used to determine the extent of internal versus external mixing by calculating the mixing state index ($\chi$). The aerosol

population was more externally mixed than internally mixed during all time periods analysed. Accumulation mode aerosol ranged from mostly internally mixed during SOA periods to mostly externally mixed during dust periods. Supermicron aerosol were most externally mixed during SOA time periods, when more SOA particles added a distinct supermicron class, and more internally mixed when dominated by a single particle type (e.g. SSA or dust). These results emphasize that neither external nor

internal mixtures fully represent the mixing state of atmospheric aerosols, even in a rural, forested environment, which has important implications for air quality and climate modelling.

## 1 Introduction

The southeastern United States has experienced neutral to cooling shifts in regional climate over the past century (Portmann et al., 2009; Saxena and Yu, 1998), in contrast to warming observed in the rest of the

United States. This has been attributed to increased formation of secondary organic aerosol (SOA) with largely cooling effects due to efficient light scattering and activity in cloud formation (Goldstein et al., 2009; Portmann et al., 2009). Regionally, the main SOA source is oxidation of biogenic volatile organic compounds (BVOCs), followed by condensation onto preexisting particles containing sulfate, nitrate, and ammonium (Anttila et al., 2007; Boyd et al., 2015; Budisulistiorini et al., 2015b; Carlton et al., 2010;

Chameides et al., 1988; Hodas et al., 2014; Lee et al., 2010; Nguyen et al., 2015; Weber et al., 2007; Xu et al., 2015b). Most studies of aerosol climate impacts in the southeast have focused on the effects of SOA, as this region has high concentrations of organic carbon, which combined with ammonium sulfate, contribute 60-90 % of fine particulate matter ($PM_{2.5}$) (Attwood et al., 2014; Boone et al., 2015; Cerully et al., 2015; Nguyen et al., 2014). However, despite the importance of SOA, the mixing of secondary

species (SOA, sulfate, nitrate, etc.) with primary particles is not fully known, particularly for forested locations impacted by regional anthropogenic emissions. The form and extent of mixing between chemical species in individual particles, i.e. mixing state, is critical for climate-relevant properties



including light scattering, water uptake, and particle acidity (Artaxo and Orsini, 1987; Cong et al., 2010; Kunkel et al., 2012; Metternich et al., 1986; Violaki and Mihalopoulos, 2010; Xu et al., 2015a). Therefore, it is important to identify the sources of aerosol particles present in the southeastern United States, as well as their size and mixing state in order to accurately assess their impact on aerosol direct and indirect

effects (Li et al.; Moise et al., 2015; Posfai and Buseck, 2010).

Mixing state is described in terms of external and internal mixtures: an external mixture consists of particles that contain only one pure species per particle, while an internal mixture describes particles that contain equal amounts of all chemical species (Ault and Axson, 2017; Posfai and Buseck, 2010; Riemer and West, 2013). The mixing states of ambient aerosol populations are complex and can vary as

a function of size, altitude, and particle age (Fu et al., 2012; Healy et al., 2014a; Moffet et al., 2010b; Pratt and Prather, 2010). Aging, or atmospheric processing such as coagulation, condensation of secondary species, and heterogeneous reactions leads to internal mixing, while freshly emitted particles are more externally mixed (Schutgens and Stier, 2014; Weingartner et al., 1997). Here, mixing state is used to describe the distribution of chemical species in a population and is purely based on composition, not

including particle morphology or other physical properties (Ault and Axson, 2017). Although the representation of mixing state in models is still an open research question (Riemer and West, 2013), an appropriate description of mixing state is critical for modeling the optical properties (Chung and Seinfeld, 2005; Jacobson, 2001; Zaveri et al., 2010) and cloud condensation nuclei (CCN) activity of  particles (Zaveri et al., 2010).

Riemer and West (2013) introduced the mixing state index ($\chi$) to quantify aerosol mixing state. This parameterization uses single particle mass fractions of individual components to calculate the average particle-specific diversity and the bulk population diversity, from which the mixing state index can then be determined. This methodology has been applied to a handful of laboratory and field studies, to date. In the laboratory, Dickau et al. (2016) used aerosol sizing and mass instrumentation to quantify

the volatile mixing state of soot. Single particle mass spectrometry data from field studies in London (Giorio et al., 2015) and as part the MEGAPOLI campaign in Paris (Healy et al., 2014b) found mixing state was dependent both upon time of day and air mass origin. Similarly, mixing state parameters were applied to computer controlled scanning electron microscopy/energy dispersive X-ray spectroscopy



(CCSEM-EDX) and scanning transmission X-ray microscopy/near edge X-ray absorption fine structure spectroscopy (STXM-NEXAFS) during the Carbonaceous Aerosol and Radiative Effects Study (CARES) in the Central Valley, CA (O'Brien et al., 2015) and in the Amazon during the GoAmazon campaign (Fraund et al., 2017), finding changes in mixing state were associated with a buildup of organic matter

and particle clusters were less diverse at remote sites, respectively. However, additional studies are needed to quantify the chemical mixing state of aerosols, particularly for rural locations.

In this study, we analyzed individual atmospheric particles collected in a rural location influenced by regional pollution in the southeastern United States during the 2013 Southern Oxidant and Aerosol Study (SOAS) to identify their size-resolved chemical composition and mixing state. CCSEM-EDX was

used to determine the size, elemental composition, and number fraction of particles containing nonvolatile cations. STXM-NEXAFS was used to characterize chemical bonding of carbonaceous components, specifically distinguishing soot from organic carbon. Mass estimates of particle elemental composition from CCSEM-EDX were calculated using a modified version of the method from O'Brien et al. (2015) to quantify the mixing state parameters for both submicron and supermicron particles during time periods

dominated by SOA/sulfate, dust, and sea spray aerosol (SSA), respectively. Additionally, the variability in the mixing state index during these three time periods of interest, showed that submicron aerosol were most internal when SOA particles dominated the aerosol population. However, supermicron particles were most internally mixed when a single source (e.g. SSA or dust) dominated the population and most externally mixed when SOA was present, along with SSA and dust. The variety of particle classes, varying

extent of secondary processing, and diverse chemical mixing states at this rural, forested site may impact climate-relevant properties of aerosols in the southeastern United States.

## 2 Experimental

### 2.1 Aerosol sample collection

Samples of atmospheric particles were collected at the SOAS Centreville, AL site (32.9030 N, 87.2500

W, 242 m AMSL) between June 5 and July 11, 2013 (Bondy et al., 2017b; Hidy et al., 2014). The site was located in a rural forested region near Talladega National Forest, at a location that is part of the



SouthEastern Aerosol Research and Characterization Network (SEARCH). Meteorological and filter sample data analyzed from the SEARCH network were used to aid selection of samples for analysis (Figure 1 and S1). Particles were collected near ground level (1 m) using a micro-orifice uniform deposit impactor (MOUDI, MSP Corp. Model 110) sampling at 30 lpm with a $PM_{10}$ cyclone (URG Model 786)

to exclude particles larger than 10 µm. The 50 % size cut-points for the MOUDI used in this analysis had aerodynamic diameters ($D_a$) of 3.2, 1.8, 1.00, 0.56, 0.32 0.18, 0.10, and 0.056 µm (Marple et al., 1991). Throughout SOAS, particles were impacted onto Cu 200 mesh TEM grids with Carbon Type B thin film (Ted Pella Inc.) for analysis with SEM-EDX and STXM-NEXAFS. Substrates in the MOUDI were collected daily from 8:00-19:00 CST and 20:00-7:00 CST (with 1 hour for substrate exchange), except

during intensive periods from June 10[th]-12[th], June 14[th]-16[th], June 29[th]-July 1[st], and July 7[th]-9[th] when the sampling schedule was 8:00-11:00, 12:00-15:00, 16:00-19:00, and 20:00-7:00 CST (Table S1). Samples were collected more frequently during intensive time periods, which were determined by meteorological and gas phase concentrations (Budisulistiorini et al., 2015b). In Figure 1, the MOUDI stages analyzed using CCSEM are noted for each sample. After collection, all substrates were sealed and stored at -22°C

prior to analysis.

## 2.2 CCSEM-EDX analysis

Particles on MOUDI stages 4-11 ($D_a$ = 0.056-1.8 µm, Figure 1) were analyzed using CCSEM (FEI Quanta environmental SEM) equipped with a field emission gun operating at 20 kV and a high angle annular dark field (HAADF) detector (Laskin et al., 2006; Laskin et al., 2002; Laskin et al., 2012). The SEM was

equipped with an EDX spectrometer (EDAX, Inc.) which was used to quantify X-rays of elements with atomic numbers ≥ C (Z = 6). A total of ~34,000 particles were analyzed during time periods denoted in Table S2, which constitute a representative cross section of the campaign. CCSEM analysis captured particle physical parameters including projected area diameter, projected area, and perimeter. Projected area diameter, which is equivalent to the diameter of a circle with the same area as the particle silhouette,

is typically larger than aerodynamic diameters measured by other analytical techniques (Bondy et al., 2017a; Hinds, 1999). For a more accurate representation of particle size, projected area diameters were converted to volume equivalent diameter using a conversion factor of 0.49 for SOA/sulfate and biomass



burning particles and 0.66 for SSA, determined from atomic force microscopy (AFM) volume calculations of particles from SOAS (Tables S3 and S4). EDX spectra from individual particles were analyzed to determine the relative abundance of 14 elements: C, N, O, Na, Mg, Al, Si, P, S, Cl, K, Ca, Ti, and Fe. Note, the Cu signal in the EDX spectra is primarily due to the Cu grid from the substrate and was

not included in CCSEM-EDX analysis.

The CCSEM-EDX data sets were analyzed using k-means clustering of the elemental composition following the method described in Ault et al. (2012) using codes written in MATLAB R2013b (MathWorks, Inc.). Clusters were grouped into source-based classes by elemental composition, including mineral dust (Axson et al., 2016; Coz et al., 2009; Creamean et al., 2016; Laskin et al., 2005; Sobanska

et al., 2003), SSA (Bondy et al., 2017b; Hopkins et al., 2008; Laskin et al., 2002), SOA/sulfate (Moffet et al., 2013; O'Brien et al., 2015; Sobanska et al., 2003), biomass burning aerosol (Li et al., 2003; Posfai et al., 2003), fly ash/metals (Ault et al., 2012; Chen et al., 2012; Shen et al., 2016), biological particles (Huffman et al., 2012), and fresh soot (Li et al., 2003). Soot forms fractal aggregates of graphitic carbon (C) which contain tens to hundreds of small spherical aggregates (Li et al., 2003). However the intense

carbon signal due to the carbon film substrate made chemical identification of soot difficult, resulting in false positives from the substrate. Because of their unique morphology, the size distribution of fresh soot particles without a large, secondary organic carbon coating altering the fractal morphology, was manually determined. Then, a scaling factor based on the SEARCH network elemental carbon mass concentrations was applied to the size distribution and this factor was used in the subsequent analysis. More information

on this correction for soot can be found in the Supplementary Information (SI), specifically Table S5.

### 2.2.1 Mass calculations and mixing state parameters

Mole percent of elements analyzed using CCSEM-EDX were converted to mass fractions using the method described by O'Brien et al (2015) and detailed in the SI. Briefly, particle volumes were calculated from projected area diameters assuming the volume of a hemisphere. Particle masses were then calculated

($\mu_i$ = density x volume) assuming the following densities for each class: 1.3 g/cm$^3$ for SOA/sulfate, biomass burning aerosol, and primary biological particles (Li et al., 2016; Manninen et al., 2014; Nakao et al., 2013); 2.0 g/cm$^3$ for SSA particles (O'Brien et al., 2015); 2.6 g/cm$^3$ for dust particles (Wagner et





al., 2009); and 3.0 g/cm³ for fly ash particles (Buha et al., 2014). To calculate the mass of each element, the elemental mole percent was converted to a weight percent, which was multiplied by the estimated particle mass.

   Diversity parameters were calculated using two different methods in this work: elemental diversity
was calculated from CCSEM-EDX results similar to O'Brien et al. (2015), and mixing state parameters due to aging were calculated as described below (which use only two diversity species: the mass fraction of elements associated with externally-mixed particles and the mass fraction of secondary species). To calculate elemental diversity parameters, the mixing entropy of each particle ($H_i$) and average particle mixing entropy $(H_\alpha)$ were calculated for each particle class as described in detail by Riemer and West
10 (2013):

$$H_i = \sum_{a=1}^{A} p^a_i \ln p^a_i \tag{1}$$

$$H_\alpha = \sum_{i=1}^{N} p_i H_i \tag{2}$$

where $p_i$ is the mass fraction of particle $i$ in the population and $p^a_i$ is the mass fraction of element $a$ in particle $i$. The particle diversity $(D_i)$ was then calculated by taking the exponent of the particle-specific
entropy $H_i$, and the average particle-specific diversity $(D_\alpha)$ was calculated by taking the exponent of $H_\alpha$. $D_\alpha$ was used as an indicator of elemental diversity for each particle class: SOA/sulfate, biomass burning particles, fly ash, dust, SSA, and biological particles.

   In addition to elemental diversity, diversity parameters were calculated to quantify the extent of particle aging. To calculate the mixing state aging parameters for the three time periods of interest, two
final mass values were calculated: the mass of single particles in a class based on the sum of elements characteristic to that class, and the mass of secondary species. The elemental mass fractions as a function of size are depicted in Figure S2. Due to the semi-quantitative nature of the lower Z elements (Laskin et al., 2006) and substrate interferences, we excluded C, N, and O from mixing state calculations, similar to O'Brien et al. (2015) The mass associated with SOA/sulfate was solely accounted for by S (if present),
and therefore was either ignored or severely underestimated. Fresh biomass particles consisted of K and Cl, fly ash particles contained Al and Si, unreacted dust particles consisted of Na, Mg, Al, Si, K, Ca, Ti,





and Fe, fresh SSA particles contained Na, Mg, Cl, K, and Ca, and biological particles contained P, Cl, and K. As a metric for aging, all sulfur was assumed to be secondary within particles, though trace primary sulfur is present in SSA and may be in other classes. Thus, for the purposes of this mixing state analysis each particle contained between one and two components: a primary source-based composition and secondary aging due to sulfur. Using the mass fractions of only these two components, $H_i$, $H_a$, and the population bulk mixing entropy $(H_\gamma)$ were calculated for each particle class

$$H_\gamma = \sum_{a=1}^{A} -p^a ln p^a \qquad (3)$$

where $p^a$ is the mass fraction of element $a$ in the population. The bulk population diversity $(D_\gamma)$ was then calculated by taking the exponent of $H_\gamma$. The mixing state index $(\chi)$ is then defined as

$$\chi = \frac{D_\alpha - 1}{D_\gamma - 1} \times 100 \qquad (4)$$

where $\chi$ can range from 0 % for an external mixture to 100 % for an internal mixture.

**2.3 STXM-NEXAFS analysis**

STXM-NEXAFS measurements of two MOUDI samples (stage 9, 50 % size cut-point of 0.18 μm, 100-200 particles analyzed per sample), June 10 and July 7, 2013 were performed at the carbon K-absorption edge (280-320 eV) to characterize chemical bonding of carbonaceous components, specifically distinguishing soot from organic carbon. STXM was conducted at the Advanced Light Source at Lawrence Berkeley National Laboratory on beamline 5.3.2. The operation of the microscope has been explained in detail by Kilcoyne et al. (2003). The software programs Matlab and Axis 2000 were both used for spectral analysis of the STXM-NEXAFS data as described by Moffet et al. (2010a); (Moffet et al., 2016). Stacks of images taken at sequentially increasing photon energies were used to obtain spatially resolved spectroscopic data at the carbon K-edge. For organic identification, pixels were identified where the post-edge minus the pre-edge (optical density (OD) at 320 eV minus OD 278 eV) was greater than



zero. For the inorganic component, particles with a ratio of the pre-edge to the post-edge (OD 278 / OD 320) greater than 0.5 were identified. To identify soot inclusions within particles, individual pixels of STXM images were analyzed and if a pixel contained 35 % or greater C=C, a peak which was identified as soot using graphitic carbon as a standard, then that pixel was identified as a soot region. Additional

details on identification of SOA-containing soot inclusions are provided in the SI.

## 3 Results and discussion

### 3.1 Descriptions of particle classes at SOAS

Although SOAS took place in a rural, forested region, a variety of particle classes were observed, and SOA/sulfate was not always the dominant individual particle class. Based on the chemical composition,

seven main particle classes were identified: SOA/sulfate, biomass burning particles, soot, and fly ash, which are typically present in the submicron (< 1 μm) regime, and mineral dust, SSA, and primary biological particles with characteristic sizes > 1 μm. Figure 2 shows SEM images of representative particles from each class and their corresponding EDX spectra. SOA/sulfate particles were identified by the elemental composition of C and O, along with either S, N, or both S and N. As all SOA particles

contained inorganic species (e.g. ammonium sulfate) in addition to organic carbon (based on STXM $OD_{pre}/OD_{post} = 0.5$, ~20 % by weight), this class is referred to as SOA/sulfate. EDX spectra of SOA/sulfate particles on Si wafers (Figure S3) confirmed C and O in the particles, as did STXM, as a check due to the carbon film substrate on TEM grids that contributes to the signal for C and O in the CCSEM-EDX analysis. The presence of S and/or N in addition to C and O is likely $NO_3^-$ or $SO_4^{2-}$, based on Raman

microspectroscopy (Craig et al., 2017), or organonitrate or organosulfate compounds, which are ubiquitous in the southeastern U.S. (Ayres et al., 2015; Bondy et al., 2018; Carlton et al., 2009; Froyd et al., 2010). SOA/sulfate particles were typically circular (circularity ranging from 0.95-1, where 1 is perfectly circular, equation in SI), though some SOA/sulfate exhibited liquid-liquid phase separation (LLPS), such as core-shell or more complex morphologies, which will be explained in a future

publication. Biomass burning aerosol particles were often circular as well (0.96 circularity), with high concentrations of K and frequently S and Cl, in addition to C and O (organic carbon) (Li et al., 2003;



Posfai et al., 2003). Fly ash particles were identified primarily by their spherical morphology (0.93 circularity) since fly ash is generated through high temperature processes (Ault et al., 2012; Chen et al., 2012), in addition to high EDX signals from O along with either Si or Al, likely in the form of $SiO_2$ or $Al_2O_3$ respectively. A final class comprised primarily of submicron particles was soot. Fresh soot particles

were identified primarily by their morphology consisting of agglomerated spheres, which had substantially smaller diameters than fly ash (Li et al., 2003). However, fresh soot was not very prevalent at SOAS and was typically present within other particles such as SOA, which will be detailed below.

In addition to submicron classes, three classes of particles predominately in the supermicron size range were observed at SOAS. Dust particles were identified by strong signals from O, Al, and Si

(aluminosilicates), often along with other elements such as Na, Mg, K, Ca, Ti, and Fe (Coz et al., 2009; Laskin et al., 2005; Sobanska et al., 2003). EDX spectra of SSA particles contained a strong Na signal ($Na^+$) and weaker Mg signal ($Mg^{2+}$) in a ~10:1 ratio, as found in seawater (Pilson, 1998), small contributions from K ($K^+$) and Ca ($Ca^{2+}$), and counter-ion elements such as N, O, S, or Cl, ($NO_3^-$, $SO_4^{2-}$, or $Cl^-$), depending on whether the SSA was fresh or aged (Bondy et al., 2017b). Finally, biological

particles typically contained primarily C due to organic macromolecules, along with lesser amounts of N (likely in the form of amines/amino acids), O, P ($PO_4^{3-}$) , and K ($K^+$) (Huffman et al., 2012), as seen in Figure 2. Overall, numerous particle classes were detected at SOAS using CCSEM-EDX based on unique chemical composition, morphology, and size.

STXM-NEXAFS was used to investigate carbonaceous particles since these particles were

prevalent at SOAS. The carbon K-edge was probed using this technique, and high spatial resolution information was obtained regarding $sp^2$ C inclusions within SOA/sulfate, which were identified as soot at 285 eV (Moffet et al., 2010a). In two samples analyzed using STXM-NEXAFS, 6.9 % and 9.9 % of particles by number contained $sp^2$ C inclusions, suggesting that a small fraction of SOA/sulfate contained soot. In comparison, organic carbon/elemental carbon (OC/EC) bulk analysis by the SEARCH network

detected ~2 % elemental carbon by mass, suggesting that although little soot was present overall, a sizeable fraction was present as small inclusions within SOA/sulfate. It is important to consider the mixing state of aerosols when modelling radiative forcing in the region, because internally mixed particles behave differently than pure components. For example, soot coated with secondary organic material may have



an enhanced absorption compared to fresh soot or soot-less SOA, though recent work has suggested that optical properties of coated soot are challenging and non-linear (Healy et al., 2015; Moffet et al., 2009; Ramanathan and Carmichael, 2008; Zhang et al., 2008). These spectra highlight that although seven main particle classes were identified, many of the particles, such as SOA and soot, were partially internally

mixed.

### 3.2 Chemical diversity observed within particle classes

Using SEM-EDX elemental mapping, morphology and the spatial distribution of species within individual particles were examined. In Figure 3, particles (a-d) were identified as dust based on their morphology and elemental composition. However, only (b-d) are aluminosilicate dust particles; (a)

contains high concentrations of Ca and S instead. Based on its chemical composition, this dust particle is most likely gypsum ($CaSO_4 \cdot 2H_2O$) (Hashemi et al., 2011). The elemental map highlights that elements present within the dust class are not homogeneously distributed among all particles. Rather, the dust class consists of externally mixed particles with varying compositions. In addition to dust, two other particle classes are represented in the elemental map in Figure 3. Particles (e-f) were identified as aged SSA due

to the high concentration of Na and Mg along with S and N (likely $SO_4^{2-}$ and $NO_3^-$) (Bondy et al., 2017b), and particle (g) is a primary biological particle, possibly coagulated with a small calcium oxide particle based on the morphology and distinctly different elemental compositions of the two components. In addition to differentiating particles among the seven particle classes identified, SEM-EDX mapping allowed investigation into whether coagulation or chemical aging of particles has occurred within

particles due to the presence of localized regions of elements or surface-layer coatings (Conny and Norris, 2011). As seen in Figure 3, very few of the particles have a homogeneous distribution of elements, though vacuum analysis and drying can modify the internal distribution of species within particles. Rather, Na, Ca, S, and Cl often appear in only a few distinct regions within particles, which can likely be attributed to heterogeneous reactions, limited diffusion, or other non-ideal behavior. The aluminosilicate dust

particle (b) in particular, has localized regions of Ca and S ($SO_4^{2-}$) on the edges of the particle, signifying that this particle has undergone aging, resulting in a more diverse physicochemical mixing state (Ault and Axson, 2017). Complex mixing states like this have been observed previously for SSA and dust, showing





that these classes of particles can be externally mixed or have surface coatings and inclusions leading to internal mixing, thereby altering their physical and chemical properties (Deboudt et al., 2012; Fitzgerald et al., 2015; Gantt and Meskhidze, 2013; Kandler et al., 2011; Kim and Park, 2012; Sobanska et al., 2012; Sullivan et al., 2007; Sullivan et al., 2009).

5       To probe the chemical diversity of each particle class, Figure 4 shows the average EDX elemental percentages for each particle class. The digital color histogram height shows the number fraction of particles in a class containing a specific element, while the color represents the mole % of the element. For example, 100 % of SOA/sulfate by number contain between 50-100 % C (mole %). To quantify elemental diversity of particles, $D_\alpha$, representing the average number of elements within particles in each

class, was calculated. $D_\alpha$ ranges from 1 (when a particle contains only one element) to $A$ number of elements. Note, due to interference from the substrate or detector, C, N, and O were not included in $D_\alpha$ calculations. CCSEM-EDX results suggest that SOA/sulfate particles were elementally the least diverse, as the primary quantifiable element was S, leading to a $D_\alpha = 1.00$. However, other studies from SOAS that used an aerosol mass spectrometer (AMS) (Guo et al., 2015; Xu et al., 2015c) or ultra-performance

liquid chromatography/electrospray ionization high-resolution quadrupole time-of-flight mass spectrometry (UPLC/ESI-HR-QTOFMS) (Budisulistiorini et al., 2015a; Riva et al., 2016) discovered that a wealth of sources contribute to SOA, resulting in hydrocarbon-like organic aerosol, isoprene-derived organic aerosol, as well as more-oxidized and less-oxidized oxygenated organic aerosol. Their analyses also showed that $SO_4^{2-}$ is the most abundant component aside from organic carbon, with significant

concentrations of $NH_4^+$ followed by $NO_3^-$, consistent with our observations of 92 % of SOA by number containing S and 68 % containing N (mole %). (Budisulistiorini et al., 2015a; Guo et al., 2015; Riva et al., 2016; Xu et al., 2017; Xu et al., 2015c).

      The composition of biomass burning particles was elementally more diverse than SOA ($D_\alpha =$ 1.92), with large contributions from $K^+$ (1-30 % by mole %) as well as organic carbon (C 20-100 % and

O 2-50 % by mole %). However, in addition to these three components, approximately 60 % of particles by number also contained $SO_4^{2-}$ (1-15 % S by mole %), 45% contained $NO_3^-/NH_4^+$ (1-10 % N by mole %), and 15 % by number contained 1-30 % Cl (mole %). The presence of Cl suggests that some of the biomass burning particles were fresh. However, because sulfate and nitrate, which are indicative of aging





(Li et al., 2003), were present more frequently, biomass burning particles detected during SOAS likely had sufficient time during transport for multiphase reactions to occur. The final submicron particle class, fly ash ($D_\alpha$ = 1.92), showed two distinct compositions in addition to aging: fly ash from SOAS consisted of primarily $SiO_2$, although approximately 15 % by number contained aluminum oxides with silicates.

Low concentrations of $SO_4^{2-}$ and $NO_3^-$ (1-5 % S and N by mole %) suggest acidic species, such as sulfuric and nitric acid, reacted with 25 % of fly ash by number.

Within the supermicron particle classes, a range of elemental compositions were observed for dust and SSA (Figure 4b). Dust was primarily composed of aluminosilicates (10-100 % O, 1-50 % Si, and 1-50 % Al by mole %), with minor contributions from other chemical species, including $CO_3^{2-}$ or organic

coatings (5-100 % C by mole %), $NO_3^-$ (1-10 % N by mole %), $SO_4^{2-}$ (1-10 % S by mole %), $Mg^{2+}$ (1-10 % by mole %), $K^+$ (1-5 % by mole %), $Na^+$ (1-10 % by mole %), $Ca^{2+}$ (1-10 % by mole %), and $Fe^{2+}/Fe^{3+}$ (1-30 % by mole %). The frequency of these minor elements in dust particles varied widely resulting in a high average particle species diversity ($D_\alpha$ = 4.43), with nitrate present in approximately 75 % of the dust population by number, and titanium oxides present in less than 5 %. The diversity of dust indicates

various sources and processing throughout SOAS, which likely contributed to time periods with distinct dust compositions due to wind speed, direction, and pollution levels. Allen et al. (2015) detected two high coarse nitrate events during SOAS, the first corresponding to high levels of SSA and dust, and the second primarily dust. The first event exhibited a higher percent of $Na^+$, not all of which was attributed to SSA due to the low $Mg^{2+}$ to $Na^+$ molar ratio, while the second event had a higher percent composition of $Ca^{2+}$.

Back trajectory analysis of the air mass origin during the two coarse particle events indicate that although the overall pattern in wind trajectories was similar, slight differences in wind patterns at the beginning of each event may have contributed to the observed differences in composition of the aerosol, suggesting a relatively local origin of the dust during the second event. The elements of SSA were more homogeneously distributed throughout the population than dust ($D_\alpha$ = 2.94), with 100 % of particles by

number containing C, O, and Na, 75 % by number containing S, and ~70 % by number containing > 1 % (mole %) N and Mg. SSA particles also showed various degrees of aging with respect to the anions, since chloride can be liberated through multi-phase reactions with acidic species such as $HNO_3$, $H_2SO_4$, and organic acids (Bondy et al., 2017b). Partially aged SSA comprised approximately 20 % of particles by





number, indicated by Cl⁻ (1-10 % Cl by mole %) in addition to nitrate and sulfate. Complete chloride depletion and aging by nitrate (1-30 % N by mole %) and sulfate (1-30 % S by mole %) was more ubiquitous though, with each secondary species present in ~90 % of SSA by number. A thorough discussion of the degree of reactive processing of SSA transported inland to Centreville can be found in

Bondy et al. (2017b).

Primary biological particles contained primarily organic carbon (50-100 % C and 5-20 % O by mole %) with minor contributions from $PO_4^{3-}$, $SO_4^{2-}$, $K^+$, in addition to other minor elements ($D_\alpha = 6.24$). The minor constituents (P, K, S) were not detected in all particle (20 % by number). The absence of these minor constituents from EDX spectra is likely the result of low concentrations compared to carbon, and

signal below the 1 % detection threshold. Although sulfate is typically an indication of aging by $H_2SO_4$ in aerosol particles, it is also naturally present in biological particles. Furthermore, because the sulfur signal intensity is on the same scale as the other minor constituents, it is not necessarily from secondary processes. Overall, throughout both the submicron and supermicron particle regimes, particle diversity varied, indicating sources of long and short range transport, and various degrees of aging of particles

within each class.

### 3.3 Variations in particle classes observed during key SOAS events

Three main time periods (SOA, dust, and SSA) were identified during SOAS that had distinctly different sources and processing (Figure 5). Figure 5a depicts the size-resolved chemical composition averaged over two SOA-dominated time periods (June 14-17, 2013 and July 7-11, 2013), two dust events (June 11-

13, 2013 and June 26-28, 2013), and two SSA events (June 10-11, 2013 and July 3-6, 2013), though only select MOUDI stages were analyzed for each sampling period. During each time period depicted, SOA/sulfate averaged > 60 % of accumulation mode (0.2-1.0 µm), and 2 % of the supermicron (1.0-5.0 µm) particles by number fraction. However, the number fraction of SOA/sulfate was highly variable between the SOA, dust and SSA periods. During the two periods dominated by SOA/sulfate depicted in

Figure 5a, the number fraction of SOA/sulfate reached up to 95 % in the accumulation mode and up to 70 % of supermicron particles. Because Centreville, AL is a forested site and BVOC emissions, such as isoprene, are high in this region, it is not surprising that SOA/sulfate dominated throughout the majority



of the campaign, particularly at small particle sizes. However the fraction of SOA/sulfate > 1 µm is noteworthy, as SOA/sulfate particles are typically considered submicron in size.

Dust was the dominant particle source during two coarse-mode nitrate events (Figure 5b) detailed previously by Allen et al. (2015) and defined more narrowly herein as June 11-13 and June 26-28, 2013

to differentiate from SSA transport time periods and account for available CCSEM data. During the dust-dominated time periods analyzed, dust constituted > 55 % of supermicron particles (1.0-5.0 µm) by number, but also contributed, on average, 26 % of accumulation mode particles (0.2-1.0 µm) by number.

Similar to dust, SSA contributed significantly to the overall particle population multiple times throughout the study, comprising approximately 35 % of particles, by number, analyzed during an event

in the middle of June (June 10-11, 2013), and at the beginning of July (July 3-6, 2013). Both of these SSA events were also characterized by high number fractions of dust, as observed in Figure 5c. During these SSA-rich periods, SSA particles were predominately larger than 1 µm (38 % by number), although notable contributions to accumulation mode number fractions of SSA were also observed (22 % by number from 0.2-1.0 µm). During these two events the degree of atmospheric processing varied, with a

considerable number fraction of partially aged SSA present during the second event compared to the first event, which was primarily fully-aged SSA (Bondy et al., 2017b).

**3.4 Nonvolatile cations at SOAS**

Recently, the potential for soluble nonvolatile cations such as $Na^+$, $Mg^{2+}$, $K^+$, and $Ca^{2+}$ to improve thermodynamic modelling results of aerosol acidity when included as inputs has been suggested,

assuming all species are internally mixed (Guo et al., 2017). As CCSEM-EDX can readily detect metals within individual particles, the number fraction of particles containing Na, Mg, K, Ca, and Fe at sub- and supermicron sizes during the SOA, dust, and SSA events is shown in Figure 6. In addition to these metals, Mn was detected within < 3 % particles by number during SOAS, and, given its low fraction, Mn was not included in further analysis. During all events, the number fraction of particles containing nonvolatile

cations increased as a function of particle size, with a higher number fraction of metal-containing particles at supermicron sizes (19-94 %) compared to submicron sizes (1-50 %). During all the time periods depicted, Na was present most frequently, closely followed by Mg, indicative of SSA particles. Fewer



particles contained K and Ca by comparison, and Fe was present within the lowest number fraction of particles, except for during the dust period when Fe was more frequent. The number fraction of metal-containing particles was not consistent throughout SOAS, but varied dramatically between the SOA, dust, and SSA periods. In general, particles during the dust and SSA events contained higher number fractions

of all nonvolatile cations, particularly Na and Mg. However, the variation between specific metals was largely dependent on the dominant particle class during each period.

Figure 7 focuses on the SOA time period and shows the number fraction of particles within each particle class that contains Na, Mg, K, Ca, or Fe (dust and SSA periods are shown in Figure S4). Within both submicron and supermicron particles, the nonvolatile cations within each class are consistent, though

a marginally larger number fraction of supermicron particles contained nonvolatile cations, likely due to detection limits for smaller particles. Less than 5 % of SOA/sulfate particles by number contained any Na, Mg, K, Ca or Fe. Conversely, all other particle classes contained metals within a substantial number of particles. Specifically, all biomass burning particles contained K and fly ash most frequently contained Na, though most fly ash contained Al or Si instead (Figure 4). Additionally, a considerable fraction of

dust particles contained Na, Mg, K, Ca, or Fe, all SSA contained Na and many contained Mg, and most primary biological particles contained Na, Mg, and K. Comparing the nonvolatile cations within each class during the SOA, dust, and SSA periods (Figures 7 and S4), the number fractions of metal-containing particles are consistent for each particle class, suggesting than an internal mixing assumption for nonvolatile cations and their presence in SOA/sulfate particles does not reflect overall particle

composition.

### 3.5 Particle aging

In contrast to nonvolatile cations, the contribution of secondary components within each class varied drastically throughout SOAS. Due to atmospheric reactions and transport of gases from nearby cities,

many of the particles analyzed from SOAS were likely not "fresh" from their source, but had undergone secondary processing by species such as $HNO_3$, $SO_2/H_2SO_4$, or organic acids. Secondary processing of particles is important because changing their chemical composition can impact light scattering and CCN



properties (Chang et al., 2010; Chi et al., 2015; Ghorai et al., 2014; Giordano et al., 2015; Hiranuma et al., 2013; Lu et al., 2011; Moise et al., 2015; Robinson et al., 2013; Sedlacek et al., 2012; Tang et al., 2016). As the chemical composition of particles varied over time, the mixing state index was used to quantify the degree of aging. The degree of secondary processing for each particle class was calculated

as the average mass fraction of sulfur per particle, see Figure S5 and details in the SI. Only sulfur was used as an indicator of aging in this study since carbon had interference from the background and nitrogen is only semi-quantitative with CCSEM-EDX (Laskin et al., 2006).

From STXM-NEXAFS, we know that most SOA particles are mixtures of organic and inorganic (mostly ammonium sulfate) components and there are almost no externally mixed organic or ammonium

sulfate particles present. As such, based on elemental composition shown in Figure 4 and the fact that C, O, and N could not be quantified in this study, particles containing only S in our mixing state analysis are presumed to be SOA/sulfate. A large fraction of sulfur in SOA was likely in the form of sulfate, since sulfate was identified as the major inorganic component within SOA (24 % wt. in fine aerosol) (Budisulistiorini et al., 2015b). Additionally, IEPOX-derived organosulfates and other organosulfates,

known to contribute to the organic aerosol fraction in Centreville (Bondy et al., 2018; Boone et al., 2015; Budisulistiorini et al., 2015b; Riva et al., 2016; Xu et al., 2015b), also contributed to the EDX sulfur content of SOA. The other five particle classes contained substantially less sulfur than SOA; SSA (20-30 wt. % S), biomass burning particles (15-25 wt. % S), dust (5-15 wt. % S), fly ash (2-10 wt. % S), and biological particles (15-25 wt. % S). SSA and biomass burning particles are both readily aged by sulfuric

acid forming $Na_2SO_4$ and $K_2SO_4$, respectively (Chen et al., 2017; Hopkins et al., 2008; Li et al., 2003), although up to 8 % of sulfate in SSA may have marine origins (Pilson, 1998). Aluminosilicate dust, the most common type of mineral dust detected at SOAS, is also aged by sulfuric acid (Perlwitz et al., 2015; Song et al., 2007; Sullivan et al., 2007). Fly ash detected at SOAS did not contain much sulfur, indicating that it was relatively fresh, or was aged more by other species such as organics, relative to sulfuric acid

(Li et al., 2017). Primary biological particles also contained low mass fractions of sulfur. However, as heterogeneous chemistry of this class of particles has not been explored as extensively as the other classes and the sulfur mass fractions did not follow the same trends for the three time periods (Figure S5), the




sulfur content in biological particles may have been, but was not necessarily the result of aging (Estillore et al., 2016).

In addition to differences in aging by sulfur for each particle class, the average mass fraction of sulfur within each class varied during the SOA-rich, dust-rich, and SSA-rich time periods (Figure S5).

Specifically, the average mass fraction of sulfur was significantly higher during the SOA-dominated time period compared to the dust and SSA periods at the 95 % confidence interval for all particle classes aside from biological (Table S6-S7). However, the mass fraction of sulfur was not statistically different between the dust and SSA periods for any particle classes (Table S8). Stagnant air masses, indicated by slower average wind speed at Centreville during the SOA/sulfate period ($1.62 \pm 0.72$) compared to the dust period

($2.34 \pm 0.95$) and SSA period ($2.10 \pm 0.97$), may have led to more aging during the SOA events.

**3.6 Quantification of mixing state using aging diversity measures**

To quantify the differences in aging during three described events, the mixing state index was calculated for the SOA, dust, and SSA time periods (Figure 8). Because the particle classes present at sub- and supermicron sizes vary dramatically, mixing state indices were calculated separately for the two size

ranges. From calculations of the average particle diversity and the bulk diversity (Figure 8a), the mixing state index, a ratio measuring how close the population is to an external or internal mixture, could be determined for each time period (Figure 8b). The mixing state indices for supermicron particles were generally the highest ($\chi$ = 19 %, 15 %, and 11 % during the SSA, dust, and SOA periods, respectively), signifying that supermicron particles were less diverse than submicron particles. The supermicron SOA

period, is more externally mixed than the SSA or dust periods for two reasons 1) because it contains the most individual particle classes (largest bulk diversity, ~5) and 2) the particle-specific diversity is highest as well, indicating that this period has a lot of aging by sulphur (and likely organic carbon), contributing to the relatively high mixing state index. The mixing state index for accumulation mode particles during the SOA and SSA periods were comparable ($\chi$ = 10 % and 9 %, respectively), while during the dust time

period many more elements were present in separate particles, leading to the most external mixture observed. The SSA time period had a more external accumulation mixing state index than the supermicron mode, which is logical as SSA dominated the supermicron during SSA periods, but the accumulation



mode still had distinct classes (e.g. SOA/sulfate). For the SOA time periods, the accumulation mode had the lowest bulk diversity and particle-specific diversity, while the supermicron mode had both high particle diversity and greater particle-specific diversity, which led to them having similar mixing state indices. Overall, Figure 8a demonstrates that time periods with low bulk diversity, which contain fewer

particle classes, have mixing state indices closer to 100 % (more internally mixed).

Mixing state indices of this work are lower than previous reports by Fraund et al. (2017) and O'Brien et al. (2015) ($\chi > 80$ % and $\chi > 40$ %, respectively), who used CCSEM-EDX and STXM-NEXAFS to analyze particles collected in the Amazon and Central Valley, CA. In both studies, calculations using STXM-NEXAFS resulted in low diversity and high mixing state indices, likely due to the inclusion of

organic carbon, which increased particle homogeneity. O'Brien et al. also calculated mixing state indices using solely CCSEM-EDX, similar to our study. From this method, O'Brien found lower mixing state indices using CCSEM-EDX ($\chi = 41\text{-}90$ %) compared to mixing state index calculations from STXM-NEXAFS results ($\chi > 60$ %). However, the mixing state index in this previous work increased during high SSA periods and periods characterized by increased mass fractions of K, Ca, Zn, and Al, suggesting that

periods with higher average particle-specific diversity were more homogeneous, simply because they contained more elements than periods dominated by carbonaceous particles. To improve upon this inherent challenge associated with quantifying mixing state using CCSEM-EDX, in the current study we calculate mixing state based on the number of particle classes and secondary species (in this study, sulfur) rather than the number of elements within particles. Quantifying mixing state parameters using this

approach is consistent with our concept of atmospheric aging since the mixing state index increases as bulk diversity decreases and the mass fraction of secondary species increases, signifying aging increases the degree of internal mixing in a population. This method of quantifying aerosol mixing state using single particle methods can be used to show the varying impact of sources and aging between different air masses at the same location.

**4 Conclusions**

Even at rural locations, a variety of particle classes with complex chemical mixing states can contribute to the aerosol population, impacting climate direct and indirect effects. During the SOAS field campaign



in Centreville, Alabama, CCSEM-EDX analysis identified the following particle classes: biological, mineral dust, SSA, fly ash, biomass burning aerosol particles, SOA/sulfate, and fresh soot. Although SOA/sulfate dominated the overall measured aerosol population, especially in the accumulation mode (0.2-1.0 µm), it was found to be present at supermicron sizes as well. Additionally, while biological

particles, mineral dust, and SSA dominated the supermicron regime, mineral dust and SSA were also observed as significant particle fractions in the accumulation mode. While some of the particle classes indicate nearby regional sources, such as fly ash transported from nearby cities and SOA/sulfate formed from the interaction of biogenic VOCs and anthropogenic pollutants, other classes point toward longer range transport, such as SSA transported from the Gulf of Mexico. From the single particle chemical

analysis conducted, complex chemical mixing states of particles with secondary processing by sulfur were observed. Finally, even though the sampling site in Centreville was located in a relatively remote region, long and short range transport of particles was evident based on not only the wide variety of particle classes and degrees of aging within each class, but also on the variation in number concentration over time. The average mass fraction of sulfur within every particle class, aside from primary biological

particles, was greater during more stagnant conditions, leading to more internally mixed particle populations.

These findings quantify the diversity in particle composition and their mixing states characteristic of the southeastern United States, suggesting that many factors and classes of particles beyond SOA/sulfate contribute to the atmospheric aerosol in this region. Submicron mineral dust and SSA may

be underappreciated sources of CCN in this region, as they are sometimes not considered when non-refractory or soot particles are the focus of measurements. Although not highly prevalent, ~8 % of SOA by number were found to contain soot inclusions, indicating that some SOA/sulfate may actually absorb in addition to scattering solar radiation, a factor which needs to be considered to accurately model radiative transfer. Additionally, since most of the particles in this region have been chemically aged with

sulfuric acid/$SO_2$, their hygroscopicity and propensity to form CCN will be altered compared to their fresh counterparts. With this information detailing the particle classes and the mixing states during SOAS, further studies can be conducted and inputs for models can be generated to more accurately assess effects of aerosols on climate in this unique region.



*Acknowledgements:* This work was supported by startup funds from the University of Michigan. The authors wish to acknowledge EPA (R835409) and NSF (AGS1228496) awarded to collaborators Prof. Steve Bertman (Western Michigan University), Prof. Paul Shepson (Purdue University), and Prof. Kerri Pratt (University of Michigan) for providing funding and logistics support for Ault Group SOAS

sampling. The CCSEM-EDX research was performed at the Environmental Molecular Sciences Laboratory (EMSL), a science user facility located at the Pacific Northwest National Laboratory (PNNL), sponsored by the Office of Biological and Environmental Research of the U.S Department of Energy. PNNL is operated for DOE by Battelle Memorial Institute under contract number DE-AC06-76RL0 1830. Travel funds to PNNL were provided by UM Rackham Graduate School and the UM Office of the

Provost. The STXM/NEXAFS particle analysis was performed at beamline 5.3.2 at the Advanced Light Source (ALS) at Lawrence Berkeley National Laboratory. Additionally, we thank Ann Marie Carlton (Rutgers University, now UC-Irvine), Lindsay Yee (UC-Berkeley), Allen Goldstein (UC-Berkeley), and Jason Surratt (UNC-Chapel Hill) for organizing SOAS and filter sampling and Dr. Manelisi V. Nhliziyo for assistance collecting samples. Karsten Baumann and the SOAS ground team at Centreville are

acknowledged for logistical assistance.

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



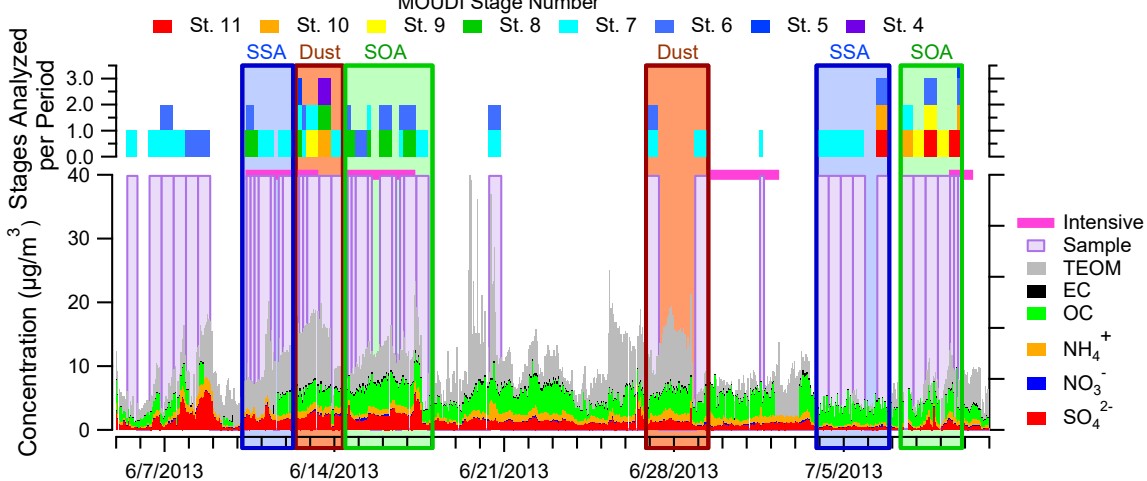

**Figure 1.** SEARCH filter sample data for Centreville, AL during SOAS with purple boxes overlaid for time periods in which CCSEM was run, and the corresponding MOUDI stages that were analyzed. SOA-rich periods denoted with green boxes (June 14-17 and July 7-11, 2013) were studied by Xiong et al. (2015), Pye et al. (2015), Xu et al. (2015c), Hu et al. (2015), and

5   Rattanavaraha et al. (2016); dust-rich periods marked with brown boxes (June 12-13 and June 26-28, 2013) were identified by Allen et al. (2015); and SSA-rich periods marked with blue boxes (June 10-11 and July 3-6, 2013) were identified by Bondy et al. (2017b). Time periods without samples analysed are due to sample damage and identification of mutually exclusive time periods. Based on periods identified in previous studies, only 29% of SOA-dominant periods, 40% of dust periods, and 51% of SSA periods were analysed here.

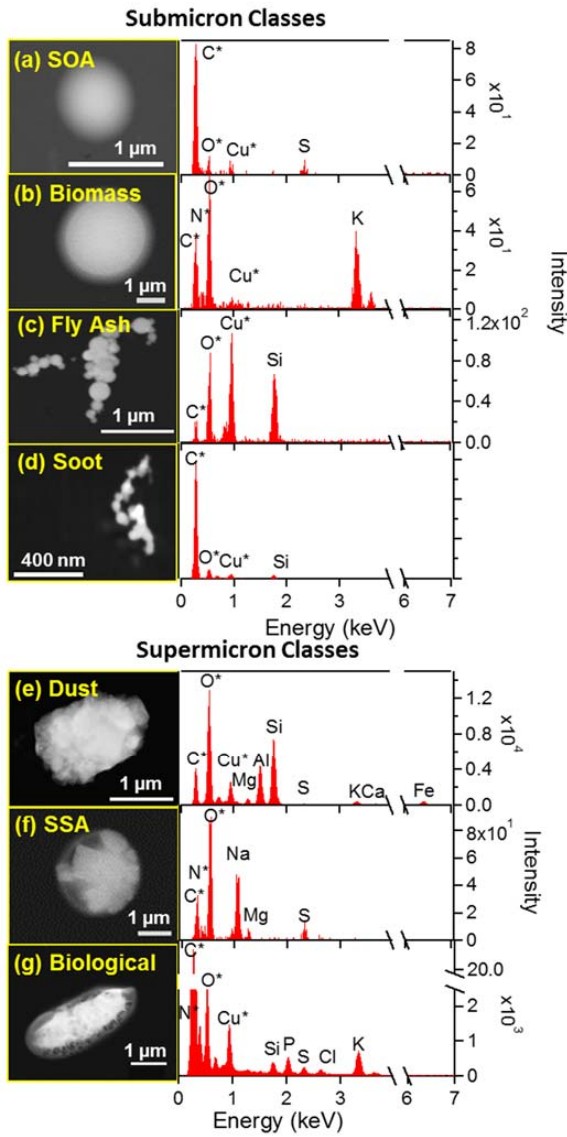

**Figure 2.** SEM images and corresponding EDX spectra for each of the main particle classes identified during SOAS within the submicron: (a) SOA, (b) biomass burning aerosol particles, (c) fly ash, (d) soot, and supermicron: (e) dust, (f) SSA, (g) primary biological, sizes. Note the elements with an asterisk are not quantitative due to interference from the substrate or detector.

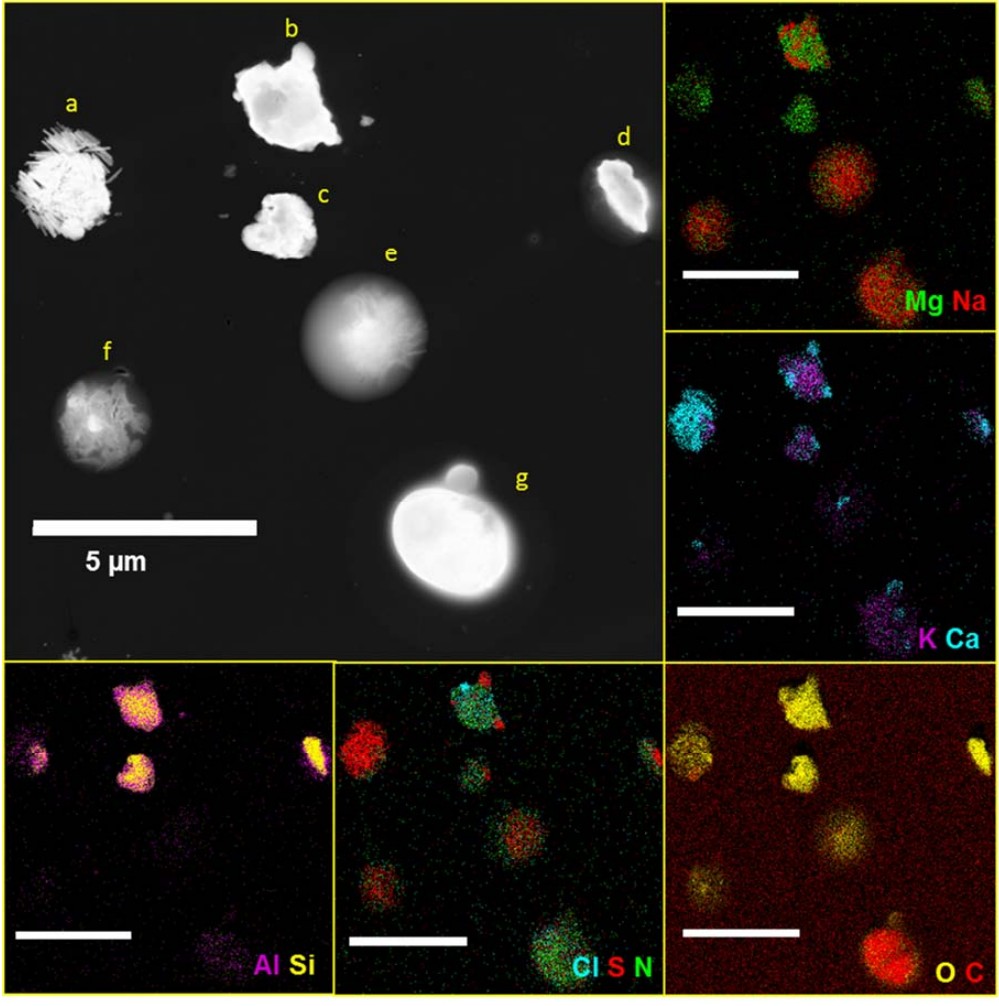

**Figure 3.** SEM image (dark field) and EDX elemental maps of particles indicated that these particle classes had various mixing states. Each of the elemental map panels corresponds to two elements overlaid to show the elemental distributions from the SEM image. The following particle classes are shown: (a-d) dust, (e-f) aged SSA, (g) primary biological.





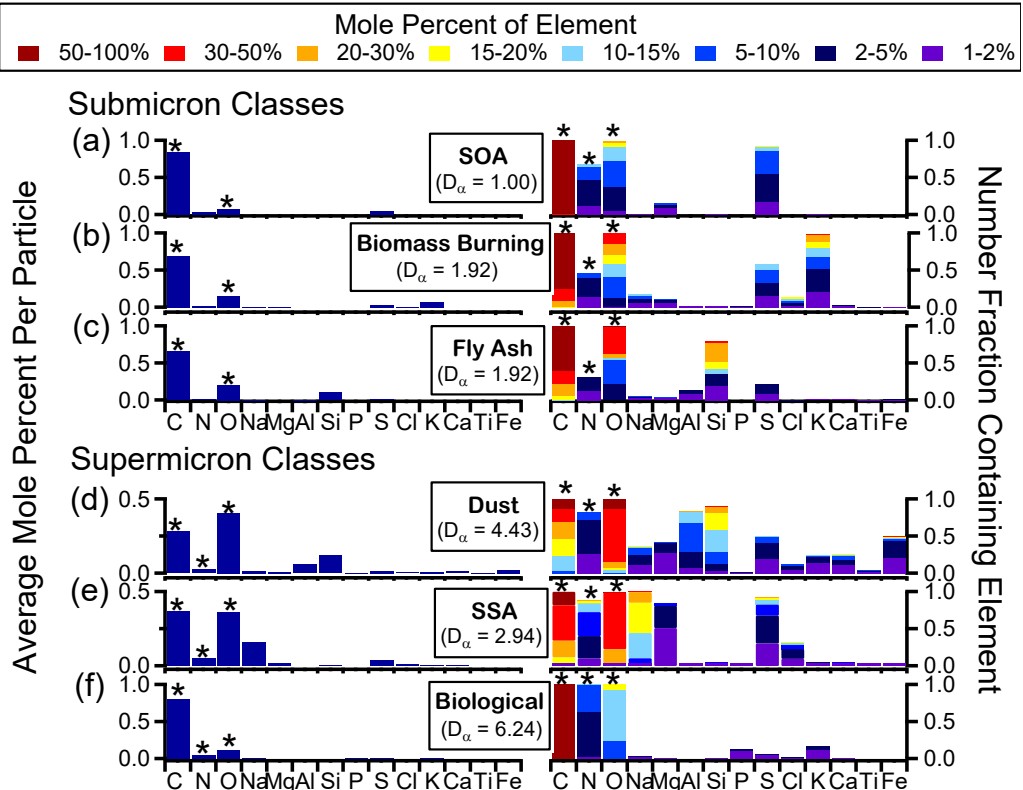

**Figure 4.** Average histograms and digital colour histograms of different particle classes from SOAS: (a) SOA, (b) biomass burning, (c) fly ash, (d) dust, (e) SSA, and (f) primary biological. Average spectra are shown on the left as the average mole percent of each element analysed by CCSEM-EDX (C, N, O, Na, Mg, Al, Si, P, S, Cl, K, Ca, and Fe). On the right, the digital colour histogram heights represent the number fraction of particles containing a specific element, and the colours represent the mole percent of that element. The average particle specific diversity ($D_\alpha$), representing the average number of elements in each particle, is calculated for each submicron and supermicron class. Note the elements with an asterisk are not quantitative due to interference from the substrate or detector and are not included in $D_\alpha$.





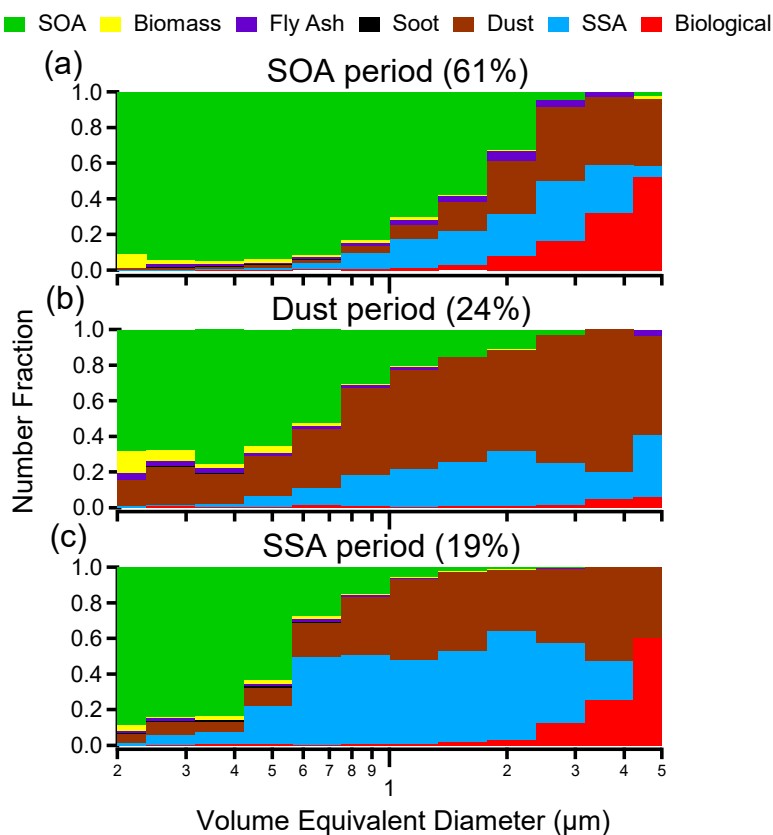

**Figure 5.** Size distributions for specific particle-rich time periods: (a) SOA-rich periods (June 14-17 and July 7-11, 2013), (b) dust-rich periods (June 12-13 and June 26-28, 2013), and (c) SSA-rich periods (June 10-11 and July 3-6, 2013). SOA periods were dominant throughout the times when samples were analyzed during SOAS (61%), high dust periods (24%) (Allen et al., 2015), and SSA periods (19%) of the time. SSA period was defined in (Bondy et al., 2017). *Literature-identified SSA periods and dust periods overlap from 6/11/2013-6/13/2013, thus the percentage of SOA, dust, and SSA periods is greater than 100% due to double counting of that time period. Only particles with volume equivalent diameters between 0.2 - 5 μm are shown due to too few particles present at larger sizes for statistical analysis.



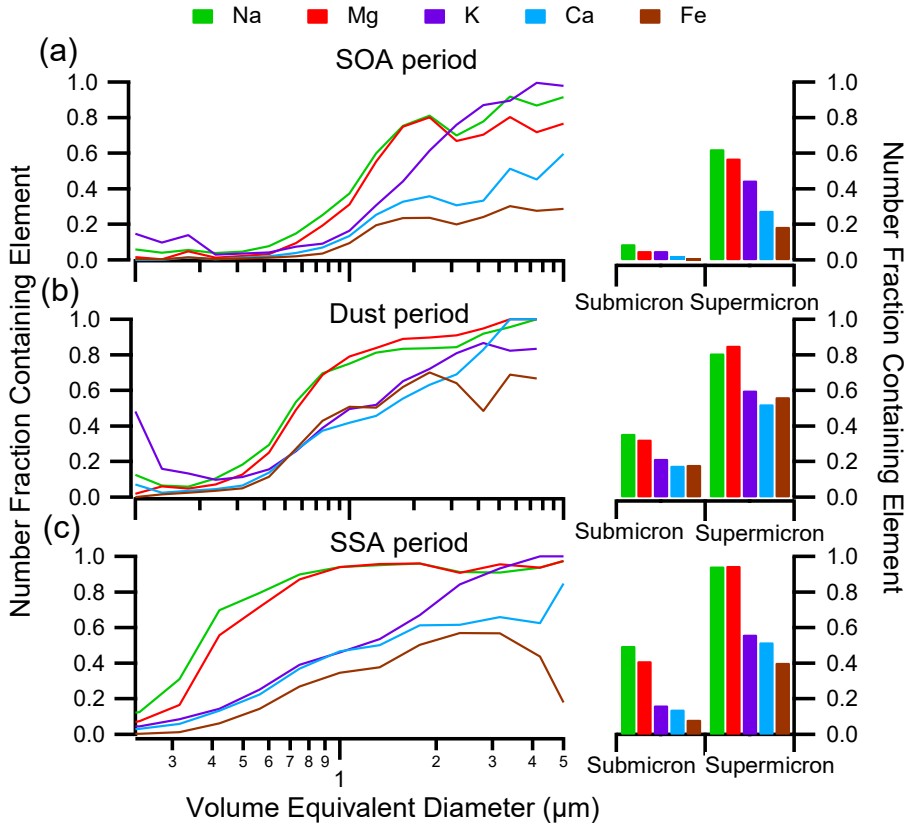

**Figure 6.** (Left) Size-resolved compositions indicate the number fraction of particles containing non-volatile cations Na, Mg, K, Ca, and Fe during the (a) SOA period, (b) dust period, and (c) SSA period. (Right) The number fraction of submicron and supermicron particles during each period containing each non-volatile cation.



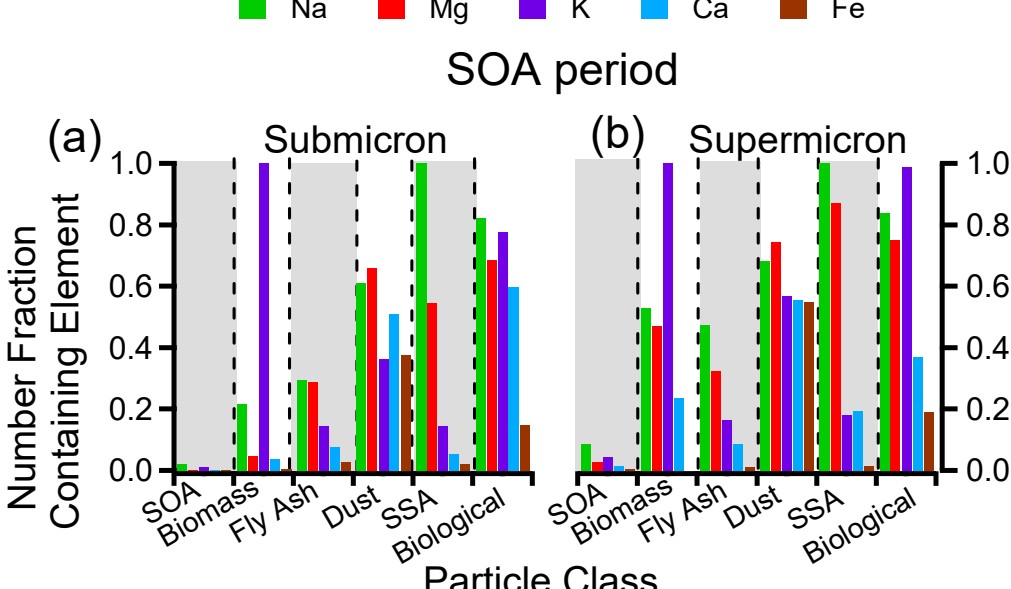

**Figure 7.** Size-resolved particle class compositions indicate the number fraction of particles in each class containing non-volatile cations Na, Mg, and Fe during the SOA period in the (a) submicron and (b) supermicron size range.





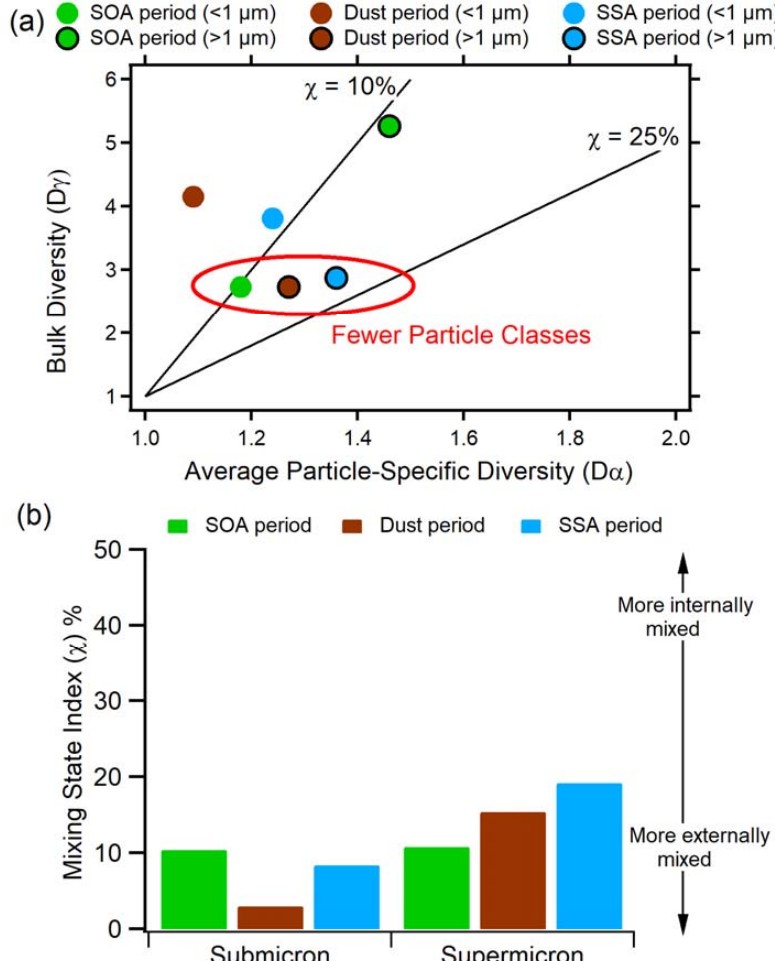

5   **Figure 8.** (a) Mixing state diagram showing the bulk diversity and average particle-specific diversity and (b) mixing

state indices for sub- and supermicron particles during the SOA, dust, and SSA periods. For submicron particles,

contributions by different sources impact mixing state.