# Peer review of "The Diverse Chemical Mixing State of Aerosol Particles in the Southeastern United States"

_Atmospheric Chemistry and Physics, 2017_

## Referee Comment (RC1) · Anonymous Referee #1 · 12 Apr 2018

The authors present a rich and very valuable dataset describing the mixing state of aerosols observed during SOAS 2013. The article is well-written and the work is technically sound. The work provides important insights which are likely to have an impact on the fields of aerosol and atmospheric chemistry. I recommend it for publication in ACP after some minor revisions.

- Page 1, line 18 change 'condensation' to 'condensation or reactive uptake'

- Page 3 lines 1-2 In addition to the many reasons the authors list for why the mixing state of aerosols is important, it is also very important for aerosol chemistry. E.g. transition metal ions cannot be important for the chemistry of SOA or biomass burning aerosols if they are not internally mixed with those aerosol types.

[Figure]

- Page 4, section 2.1 Can the authors comment on how the potential loss of semivolatile organic species in the MOUDI and under vacuum may have biased the results?

- Page 6, section 2.2.1, as the authors state, a number of assumptions are required regarding particle shape and density in order to derive mass information. Please add some discussion of the uncertainties inherent in these calculations and how this is propagated to the final results.

- page 9, line 10-13 the meaning of this sentence is clear but the way that it is written is confusing, please rephrase

- Page 10, line 8 'predominantly'

- page 16 line 13 'containing'?

- I may have missed the explanation - but why is there SSA in Centerville? This seems surprising.

- Page 20, line 21 - this statement about soot inclusions making SOA light-absorbing is a bit overreaching. In a model wouldn't that particle be considered an aged soot particle rather than SOA, and therefore already be represented as absorbing? Also, the community has considered SOA to be potentially light-absorbing (brown carbon) for some time.

- page 30, Figure 1 - this figure is very hard to understand. It is problematic that some of the same colors are used to indicate MOUDI stages vs. aerosol components. Maybe this needs to be broken up into several panels or separate figures.

---

## Referee Comment (RC2) · Anonymous Referee #2 · 22 Apr 2018

The paper presents a highly detailed study on quantification of mixing states of individual aerosol particles collected during the SOAS field campaign in Centreville – an array of complimentary microscopic techniques is used along with appropriate statistical analysis across different sampling episodes and as a function of particle size and type. The paper is very well written, such detailed data sets will be of high relevance to the community and I recommend it for publication to ACP after authors address the following minor revisions:

MOUDI sampling – was it wet/dry deposition, please add information on the relative humidity during collection; in terms of the storage – could authors please elaborate on the storage conditions, in particular how samples were sealed and then frozen/unfrozen.

Fig. 2 b – biomass burning aerosol particle – size is ∼2-3 microns, not submicron –

[Figure]

perhaps select a smaller one

Fig. 3 – only supermicron classes of particles are shown – it would be valuable to add a similar figure but for submicron particles and discuss the differences/similarities SSA/aged SSA – how confident the authors are with this particle class assignment? Based on images shown in

Fig. 3 I am not sure I can see "aged" SSA, particles e-f don't appear to have noticeable amounts of Cl, which one would expect for a NaCl core.

STXM-NEXAFS results – it would be valuable to include NEXAFS Carbon K-edge spectra of representative particles (authors refer to these results throughout the paper yet do not show actual data) + chemical mapping to illustrate mixing states – in particular over similar particles as those analyzed by SEM/EDX – I would be very curious to see, if possible, same particles analyzed using these complimentary microscopic techniques

Fig 4 + corresponding text - The average particle specific diversity is calculated for each submicron and supermicron classes of particles – could authors comment how reproducible these values are? Perhaps add a standard deviation for each class?

---

## Author Comment (AC1) · 2 Aug 2018

**Reviewer: 1**

Comments: The authors present a rich and very valuable dataset describing the mixing state of aerosols observed during SOAS 2013. The article is well-written and the work is technically sound. The work provides important insights which are likely to have an impact on the fields of aerosol and atmospheric chemistry. I recommend it for publication in ACP after some minor revisions.

1. Page 1, line 18 change 'condensation' to 'condensation or reactive uptake'
   Author Response: Authors modified the text to reflect this change on **Page 2 Line 18**.

2. Page 3 lines 1-2 In addition to the many reasons the authors list for why the mixing state of aerosols is important, it is also very important for aerosol chemistry. E.g. transition metal ions cannot be important for the chemistry of SOA or biomass burning aerosols if they are not internally mixed with those aerosol types.
   Author Response: Authors modified the text to include aerosol chemistry, such as transition metal ion dissolution, as an additional motivation for better understanding particle mixing state on **Page 3 Line 2**.

3. Page 4, section 2.1 Can the authors comment on how the potential loss of semivolatile organic species in the MOUDI and under vacuum may have biased the results?
   Author Response: The authors believe that the potential loss of semivolatile organic species may have led the EDX results to suggest particles contained less organic carbon. This may have led them to appear slightly more externally mixed than they were on the atmosphere. A sentence has been added on this on **Page 5 Lines 21-24**.

4. Page 6, section 2.2.1, as the authors state, a number of assumptions are required regarding particle shape and density in order to derive mass information. Please add some discussion of the uncertainties inherent in these calculations and how this is propagated to the final results.
   Author Response: Particle projected area diameters were converted to volume equivalent diameters assuming that impacted particles were spheres, which results in hemispheres on the substrate, and a spreading ratio determined by measuring particles from SOAS with atomic force microscopy (AFM). The densities were chosen from the electron microscopy literature for each particle source categorized. The volumes were multiplied by the densities to get mass. There is certainly inherent uncertainty when making assumptions about density and shape. This uncertainty is difficult to quantify and a small portion of the overall analysis (only related to Figure 7). A statement addressing this uncertainty and its impact on mass uncertainty is now included on **Page 7, Line 13-15**.

5. Page 9, line 10-13 the meaning of this sentence is clear but the way that it is written is confusing, please rephrase
   Author Response: Authors rephrased this sentence for clarity on **Page 9 Lines 23-26**.

6. Page 10, line 8 'predominantly'
   Author Response: Authors corrected the spelling of 'predominantly' on **Page 10 Line 21**.

7. Page 16 line 13 'containing'?
   Author Response: Authors reworded **Page 16 Lines 22-24** for clarity.

8.  I may have missed the explanation - but why is there SSA in Centerville? This seems surprising.

    Author Response: Although Centreville, AL is located over 300 km from the Gulf of Mexico, Bondy et al. (2017) showed that SSA was found to contribute significantly to supermicron and accumulation mode particle concentrations during specific events during SOAS when air masses were transported from the south. This surprising finding was discussed in detail in the authors manuscript detailing SSA observed during SOAS, which built on substantially expanded upon work published in Allen et al. (2015).

    **Bondy, A. L.**, Wang, B., Laskin, A., **Craig, R. L.**, Nhliziyo, M. V., Bertman, S. B., Pratt, K. A., Shepson, P. B., and **Ault, A. P.***: Inland Sea Spray Aerosol Transport and Incomplete Chloride Depletion: Varying Degrees of Reactive Processing Observed during SOAS, Environ. Sci. Technol., 51, 9533-9542, 2017.

    Allen, H. M.; Draper, D. C.; Ayres, B. R.; **Ault, A. P.**; **Bondy, A. L.**; Takahama, S.; Modini, R.; Baumann, K.; Edgerton, E.; Knote, C.; Laskin, A.; Wang, B.; Fry, J. L. Influence of mineral dust and sea spray supermicron particle concentrations and acidity on inorganic $NO_3^-$ aerosol during the 2013 Southern Oxidant and Aerosol Study. *Atmospheric Chemistry and Physics*, **2015**, 15(18): 10669-10685.

9.  Page 20, line 21 - this statement about soot inclusions making SOA light-absorbing is a bit overreaching. In a model wouldn't that particle be considered an aged soot particle rather than SOA, and therefore already be represented as absorbing? Also, the community has considered SOA to be potentially light-absorbing (brown carbon) for some time.

    Author Response: The reviewer makes a good point about how complicated it can be to represent mixed organic and soot particles optically with respect to scattering and absorption. The sentence in question was removed and the prior sentence revised to highlight the differences that thick SOA coatings on soot could have, as well as the conversion from fractal to compact soot morphologies, **Page 21 line 9**.

10. Page 30, Figure 1 - this figure is very hard to understand. It is problematic that some of the same colors are used to indicate MOUDI stages vs. aerosol components. Maybe this needs to be broken up into several panels or separate figures.

    Author Response: The reviewer makes a good point about confusing and overlapping colors in Figure 1. The figure has been revised to reduce overlap in the color scheme and simplify the figure.

**Reviewer: 2**

Comments: The paper presents a highly detailed study on quantification of mixing states of individual aerosol particles collected during the SOAS field campaign in Centreville – an array of complimentary microscopic techniques is used along with appropriate statistical analysis across different sampling episodes and as a function of particle size and type. The paper is very well written, such detailed data sets will be of high relevance to the community and I recommend it for publication to ACP after authors address the following minor revisions.

1. MOUDI sampling – was it wet/dry deposition, please add information on the relative humidity during collection; in terms of the storage – could authors please elaborate on the storage conditions, in particular how samples were sealed and then frozen/unfrozen.

   Author Response: Particles were intertially impacted at ambient conditions without being dried prior to impaction (beyond effects from sampling via an impactor). The relative humidity for the entire field campaign is plotted in Figure S1. This figure indicates that RH varied from ~40-100% throughout SOAS (and sample collection). Additional details regarding sample storage conditions were added to the methods section on **Page 5 Lines 14-19**.

2. Fig. 2 b – biomass burning aerosol particle – size is ~2-3 microns, not submicron –perhaps select a smaller one

   Author Response: The reviewer makes a fair point. Although the $D_{pa}$ of the biomass burning particle is slightly larger than the mode of most biomass burning size distributions, it was chosen to provide the best image quality and EDX signal. In addition, though ~2-3 microns in the image, after accounting for spreading upon impaction the particle is only ~1-1.5 μm in diameter ($D_{ve}$). (See SI for details on how $D_{pa}$ was converted to $D_{ve}$).

3. Fig. 3 – only supermicron classes of particles are shown – it would be valuable to add a similar figure but for submicron particles and discuss the differences/similarities SSA/aged SSA – how confident the authors are with this particle class assignment?

   Author Response: A similar figure for a sample of submicron particle would be nice and we would like to include that as well, but EDX mapping is not as effective for most of the submicron particle classes due to smaller size and beam sensitivity. This is because dust, SSA, and primarily biological particles are relatively stable under electron beam irradiation, SOA and biomass burning aerosol are easily damaged and EDX elemental maps with sufficient X-ray signal cannot be collected for these specific particle classes.

   With regards to the SSA/aged SSA particle class assignment, authors are quite confident with the assignment of SSA, which is discussed in detail in Bondy et al. (2017). Briefly, an Na:Mg mole ratio of ~10:1 is used as a marker for SSA as this is the ratio of $Na^+/Mg^{2+}$ in seawater), which has been shown in multiple publications (Ault et al., 2013b; Prather et al., 2013). The distinction between fresh and aged SSA, noted by the presence or lack of Cl and replacement by S and/or N, was discussed in detail in Bondy et al. (2017), which utilized findings from prior studies of SSA reacted with acidic gases (e.g. $HNO_3$) (Ault et al., 2014; Ault et al., 2013a). In brief, no fresh SSA was detected at Centreville during SOAS but rather various degrees of chloride depletion were observed resulting in partially-aged and fully-aged SSA.

4. Based on images shown in Fig. 3 I am not sure I can see "aged" SSA, particles e-f don't appear to have noticeable amounts of Cl, which one would expect for a NaCl core.

   Author Response: As described in the authors' previous publication (noted above), the majority of SSA observed at have chloride depleted relative to the ratio with sodium in seawater. The particles in Figure 3 still have some chloride, as the reviewer notes, but in a lower ratio to sodium when compared to seawater. This is discussed in detail in Bondy et al. (2017)

5. STXM-NEXAFS results – it would be valuable to include NEXAFS Carbon K-edge spectra of representative particles (authors refer to these results throughout the paper yet do not show actual data) + chemical mapping to illustrate mixing states – in particular over similar particles as those analyzed by SEM/EDX – I would be very curious to see, if possible, same particles analyzed using these complimentary microscopic techniques

   Author Response: Due to limited beam-time access, the authors only have STXM-NEXAFS data for SOA and primary biological particles. We added a STXM image, chemical map, and NEXAFS spectrum for a biological particle to the Supporting Information. SOA are the focus of a companion publication. For more detailed comparisons of SEM-EDX and STXM see O'Brien et al. (2015).

6. Fig 4 + corresponding text - The average particle specific diversity is calculated for each submicron and supermicron classes of particles – could authors comment how reproducible these values are? Perhaps add a standard deviation for each class?

   Author Response: The use of particle mixing state metrics, such as diversity, is a relatively recent development (Riemer and West, 2013). These values are reproducible to within 5-10%. A standard deviation is not necessarily the best manner for representing uncertainty, but the reviewer's point is well taken. To make clear that the calculations have limited precision, the number of significant figures on Figure 4 has been reduced to account for the uncertainty and a note about uncertainty added to the caption.

**References**

Allen, H. M., Draper, D. C., Ayres, B. R., Ault, A., Bondy, A., Takahama, S., Modini, R. L., Baumann, K., Edgerton, E., Knote, C., Laskin, A., Wang, B., and Fry, J. L.: Influence of crustal dust and sea spray supermicron particle concentrations and acidity on inorganic NO3- aerosol during the 2013 Southern Oxidant and Aerosol Study, Atmos. Chem. Phys., 15, 10669-10685, 2015.

Ault, A. P., Guasco, T. L., Baltrusaitis, J., Ryder, O. S., Trueblood, J. V., Collins, D. B., Ruppel, M. J., Cuadra-Rodriguez, L. A., Prather, K. A., and Grassian, V. H.: Heterogeneous reactivity of nitric acid with nascent sea spray aerosol: large differences observed between and within individual particles, J. Phys. Chem. Lett., doi: 10.1021/jz5008802, 2014. 2493-2500, 2014.

Ault, A. P., Guasco, T. L., Ryder, O. S., Baltrusaitis, J., Cuadra-Rodriguez, L. A., Collins, D. B., Ruppel, M. J., Bertram, T. H., Prather, K. A., and Grassian, V. H.: Inside versus outside: ion redistribution in nitric acid reacted sea spray aerosol particles as determined by single particle analysis, J. Am. Chem. Soc., 135, 14528-14531, 2013a.

Ault, A. P., Moffet, R. C., Baltrusaitis, J., Collins, D. B., Ruppel, M. J., Cuadra-Rodriguez, L. A., Zhao, D., Guasco, T. L., Ebben, C. J., Geiger, F. M., Bertram, T. H., Prather, K. A., and Grassian, V. H.: Size-dependent changes in sea spray serosol composition and properties with different seawater conditions, Environ. Sci. Technol., 47, 5603-5612, 2013b.

Bondy, A. L., Wang, B., Laskin, A., Craig, R. L., Nhliziyo, M. V., Bertman, S. B., Pratt, K. A., Shepson, P. B., and Ault, A. P.: Inland Sea Spray Aerosol Transport and Incomplete Chloride Depletion: Varying Degrees of Reactive Processing Observed during SOAS, Environ. Sci. Technol., 51, 9533-9542, 2017.

O'Brien, R. E., Wang, B. B., Laskin, A., Riemer, N., West, M., Zhang, Q., Sun, Y. L., Yu, X. Y., Alpert, P., Knopf, D. A., Gilles, M. K., and Moffet, R. C.: Chemical imaging of ambient aerosol particles: Observational constraints on mixing state parameterization, J. Geophys. Res.: Atmos., 120, 9591-9605, 2015.

Prather, K. A., Bertram, T. H., Grassian, V. H., Deane, G. B., Stokes, M. D., DeMott, P. J., Aluwihare, L. I., Palenik, B. P., Azam, F., Seinfeld, J. H., Moffet, R. C., Molina, M. J., Cappa, C. D., Geiger, F. M., Roberts, G. C., Russell, L. M., Ault, A. P., Baltrusaitis, J., Collins, D. B., Corrigan, C. E., Cuadra-Rodriguez, L. A., Ebben, C. J., Forestieri, S. D., Guasco, T. L., Hersey, S. P., Kim, M. J., Lambert, W. F., Modini, R. L., Mui, W., Pedler, B. E., Ruppel, M. J., Ryder, O. S., Schoepp, N. G., Sullivan, R. C., and Zhao, D.: Bringing the ocean into the laboratory to probe the chemical complexity of sea spray aerosol, Proc. Natl. Acad. Sci. U. S. A., 110, 7550-7555, 2013.

Riemer, N. and West, M.: Quantifying aerosol mixing state with entropy and diversity measures, Atmos. Chem. Phys., 13, 11423-11439, 2013.